# Prediction of Surface Roughness of SLM Built Parts after Finishing Processes Using an Artificial Neural Network

**Daniel Soler** [1,*] , **Martín Telleria** [1] , **M. Belén García-Blanco** [2] , **Elixabete Espinosa** [2] , **Mikel Cuesta** [1] and **Pedro José Arrazola** [1]

1 Manufacturing Department, Faculty of Engineering-Mondragon Unibertsitatea, 20500 Arrasate-Mondragon, Spain; martin.telleria@alumni.mondragon.edu (M.T.); mcuesta@mondragon.edu (M.C.); pjarrazola@mondragon.edu (P.J.A.)
2 CIDETEC, Basque Research and Technology Alliance (BRTA), Po. Miramón 196, 20014 Donostia-San Sebastián, Spain; bgarcia@cidetec.es (M.B.G.-B.); eespinosa@cidetec.es (E.E.)
* Correspondence: dsoler@mondragon.edu

**Abstract:** A known problem of additive manufactured parts is their poor surface quality, which influences product performance. There are different surface treatments to improve surface quality: blasting is commonly employed to improve mechanical properties and reduce surface roughness, and electropolishing to clean shot peened surfaces and improve the surface roughness. However, the final surface roughness is conditioned by multiple parameters related to these techniques. This paper presents a prediction model of surface roughness (Ra) using an Artificial Neural Network considering two parameters of the SLM manufacturing process and seven blasting and electropolishing processes. This model is proven to be in agreement with 429 experimental results. Moreover, this model is then used to find the optimal conditions to be applied during the blasting and the electropolishing in order to improve the surface roughness by roughly 60%.

**Keywords:** surface roughness; additive manufacturing; SLM; artificial neural network; blasting; electropolishing

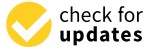



## 1. Introduction

In recent years, Additive Manufacturing (AM) is gaining importance as an alternative to conventional manufacturing processes [1], especially when they are focused on small productions and conceptual studies [2]. Concerning additive manufactured workpiece materials, titanium alloys, especially alloy Ti-6Al-4V, which represents 50% of all titanium production [3], are widely used for medical, aeronautical and automotive applications, due to their low density, combined with their strength at elevated [4] temperatures and corrosion resistance [5]. One of the main drawbacks of this technology, the poor surface quality of as-built parts, is due to powder characteristics and process parameters.

One of the parameters commonly used to characterize this surface quality is the surface roughness, and it is a fact that the surface roughness of as-built parts obtained by AM technologies is far away from that obtained by other technologies such as conventional machining processes. For instance, values of Ra = 5–40 μm for SLM and Ra = 25–130 μm for EBM were obtained for Ti-6Al-4V [6].

Moreover, geometrical features of the built part, namely the building angle of the inclination planes [7], also affect its surface roughness. The available literature includes works that have proven the connection between the building angle of an inclination plane produced by powder bed AM technologies, its surface roughness [8,9] and its mechanical properties as strength and ductility [10] or fatigue life [11]. The layered structure of powder bed technologies provokes a stair-step effect, which strongly affects the surface roughness, measured perpendicularly to the steps. Therefore, in order to ensure that an

AM product has a high-quality surface with homogenous roughness, the application of surface treatments is crucial.

Usually, post processes and surface treatments such as machining, grinding, blasting and electropolishing [12–14] are used to reduce the surface roughness of AM parts. However, blasting is the most frequently used one (i) to reduce the vast roughness of as-built samples and also, (ii) to improve mechanical properties. Electropolishing is the treatment applied when high-quality bright surfaces are required.

Blasting implies the impact of a jet of abrasive particles, usually corundum or glass microspheres, against a surface using compressed air applied with controlled pressure. These impacts provoke surface textures and roughness reduction.

Electropolishing (EP) reduces surface roughness by means of selective electrochemical dissolution. The metal to be treated is immersed in an electrolyte, which must be carefully selected for each material, anodically polarized and dissolved by applying a specific amount of current or voltage. This process was successfully applied to polish different AM materials and the resulting surfaces are usually bright, clean and smooth [15]. Moreover, EP surfaces could have additional properties such as easy cleaning and maintaining and improved corrosion resistance due to the passive oxide layer that grows on the metallic surface during the process.

The benefits of the combination of these two processes, blasting and electropolishing, were previously reported [16]. However, the optimization of both process parameters applied sequentially, also considering the initial surface state, has not been performed.

The prediction of the final roughness of the part after such processes has received little attention in the literature [17]. Analytical approaches such as the ones developed by Hashimoto and DeBra [18] or Wan et al. [19] are usually limited to considering only a small number of influencing parameters and the assumption of simplifications that could lead to poor predictability [20]. When more parameters are taken into account, especially if they are expected to have nonlinear behaviors, it is very usual to use Artificial Intelligence (AI) methods, for example, Salgado et al. [21] used least-square support vector machines to predict surface roughness for a turning process, or Li et al. [22] used six different AI methods to predict the surface roughness in extrusion-based additive manufacturing.

In the present work, the surface corresponds to Ti-6Al-4V alloy flat specimens built by Selective Laser Melting (SLM) after being finished by blasting and electropolishing techniques. Input variables to be taken into account were industrial variables considering specimens characteristics (deposition angle and as-built surface roughness), blasting parameters (type of abrasive particles, time and pressure), and the electropolishing ones (time, voltage and agitation frequency), making it very difficult to develop an analytical or a numerical model that relates them with the surface roughness [23]. Therefore, following the example of previous works such as [24–26], an Artificial Neural Network (ANN) technology, which is a branch of artificial intelligence that attempts to achieve human brain capability [27], was designed and trained to predict surface roughness (Ra). The predicted roughness of treated surfaces was found to be in good agreement with the experimental tests carried out.

Then, following the examples provided by the literature [28,29] using the developed ANN, some optimization algorithms were employed to determine the finishing conditions to be used in order to minimize the Ra given to as-built surface roughness. This provides a powerful tool to AM manufacturers, as it could help to determine parameters of finishing processes that improve the quality of obtained surfaces by roughly 60%.

The paper is organized as follows: First, are the experimental procedures which include the additive manufacturing process, blasting and electropolishing techniques details, and the theoretical background concerning surface roughness and the measuring description. Then, in Section 3, a short introduction to ANN is given, which is followed by details of the ANN design: structure, algorithms, functions and validation. The optimization algorithms are explained in Section 4. Finally, the obtained results are discussed and conclusions are drawn in the last section.

## 2. Materials and Methods

As a first approach, an ANN is an AI able to predict some issues depending on several input variables; in the present case, the ANN should predict the surface roughness of a flat Ti6Al4V specimen after being blasted and then finished by an electropolishing process. In order to precisely determine input variables of this ANN, in this section, details of the procedures of these three steps are given. Finally, there are also some details of surface roughness measurements given.

### 2.1. Additive Manufactured Specimens

Specimens were produced by SLM using a powder Ti6Al4V Gr 23 alloy (O < 1300 ppm) with spherical morphology and a particle size distribution between 20–63 μm, delivered by LPW Tech. Ltd. An MCP SLM Realizer 250 equipped with 200 W fiber laser machine was used applying laser power of 200 W, scanning speed of 900 mm s$^{-1}$, hatch space of 0.12 mm and 50 μm layer thickness. Parts with different planes were built varying the building angle from 0° to 90° with respect to the building platform.

### 2.2. Blasting Set Up

Samples were blasted using round-shape glass microspheres of 40–70 μm sometimes preceded by another blasting treatment with prismatic-shaped corundum of 90–140 μm. The process was applied in a Pallinatrici Norblast (S9 Model) grit blaster, using a nozzle diameter of 8 mm. The distance between sample holder and nozzle was set at 5 cm and the pressure drop across the nozzle was varied from 3 to 6.5 bar. Blasting treatment length was modified from 3 to 15 min.

### 2.3. Electropolishing Experimental Setup

Electropolishing experiments were carried out at constant voltage in a two-electrode system. The cathode was a Cp Ti foil and the anode was the Ti6Al4V specimen. A non-aqueous electrolyte comprising ethyl alcohol (700 mL L$^{-1}$), isopropyl alcohol (300 mL L$^{-1}$), AlCl3 (60 g L$^{-1}$) and ZnCl2 (250 g L$^{-1}$) was used. The electrolyte temperature was kept constant at 30 °C. Electropolishing time varied from 10 to 50 min and process voltage was changed from 38 V to 50 V depending on the piece. Magnetic stirring was employed at the bottom of the electrolytic cell to maintain a control hydrodynamic of the electrolyte, and it varied from 100 to 200 rpm. Subsequent to EP, substrates were rinsed with distilled water and dried with air. Table 1 shows the range of each parameter involved in the experimental data set.

**Table 1.** Complete set of parameters involved in the experimental data set and the corresponding values/range.

| | Process | Parameters | Values/Range |
|---|---|---|---|
| Manufacturing | | Deposition angle (anC) | 0, 10, 20, 30, 40, 45, 50, 60, 70, 80, 90 (°) |
| | | As-built Ra (Ra0) | [3.88,15.42] (μm) |
| Blasting | 1 | Blasting particles (G1) | Glass microsphere/Corundum |
| | | Pressure (P1) | 3, 5.5, 6.5 (bar) |
| | | Time (T1) | 3, 5, 6, 13, 15 (min) |
| | 2 | Time (T2) | 3, 6 (min) |
| Electropolishing | | Voltage (V) | 10, 17, 30, 33, 35, 50 (V) |
| | | Time (T3) | 38, 50 (min) |
| | | Magnetic stirring Frequency (A) | 100, 150, 200 (rpm) |

### 2.4. Surface Roughness Measurements

Surface roughness was measured using a Taylor-Hobson Talysurf-Intra 50 mm profilometer equipped with a 60 mm tracer, finished in a diamond tip with a radius of 2 microns and an angular opening on 90°. Owing to the inhomogeneity of the as-built surfaces, the

averaged roughness was calculated considering three measurements on x-axis and three measurements on y-axis, both on as-built samples and on treated ones. The surface roughness was calculated by the averaged Ra values (average of the distances between the roughness profile and the intermediate line of the measurement length).

To measure the profiles, a Gauss filter was used; this filter needs a cut-off (λc) that corresponds to the sampling length and that is selected based on the expected value for Ra. ISO 428/97 standard was followed, therefore, depending on the samples, a cut-off/λc = 2.5 or 0.8 mm was used. (Ls = 0.025 mm).

## 3. ANN Design and Algorithms

### 3.1. Artificial Neural Network Background

Artificial neural network (ANN), which is a computational system inspired by the biological brain, is now a popular modeling tool widely used to predict non-linear behaviors, as electropolishing and blasting techniques are. The most extended type of ANNs are those based on feedforward back-propagation networks, which approximate an unknown function $f^*(x) = y$ by defining a mapping $f(x;w,b) = y$ and learn the values of the parameters $w$ and $b$ that result in the best approximation of $f$.

These ANNs are made of multiple layers with a certain number of neurons each. In the case of fully connected ANNs, each neuron in a particular layer gets signals from all neurons of previous layers. There are three different types of layers: (i) the input layer is composed of some neurons that brings the initial data ($x$) into the system, for further processing by neurons of (ii) the hidden layers. These ones bring the information to the neurons of the (iii) output layer that gives the output value ($y$). Each neuron contains weights ($w$) and a bias ($b$), that are adjusted during the training process, and a transfer function (Figure 1). The typical transfer functions are the log-sigmoid, the tan-sigmoid or a pure linear function. A non-linear transfer function must be used in hidden layers if a non-linear relationship between input and output vectors is expected. For function fitting problems the output layer is typically a linear layer [30].

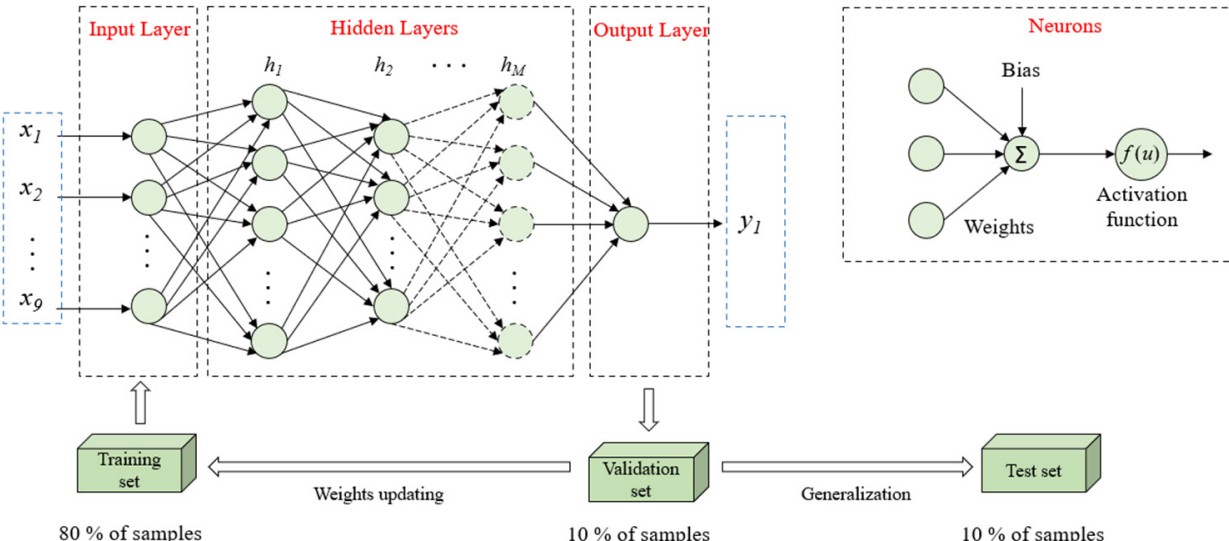

**Figure 1.** Layout for feedforward artificial neural network to predict the value of a single variable depending on 9 input parameters. Each neuron is defined by some weights, a bias and an activation function. Experimental data are used to train and validate the ANN.

Initially, all weights and biases of all neurons of the ANN are randomly defined; it is during the training process that they are adjusted to optimize the network performance, which is defined by a performance function. The training process could be implemented in incremental mode or in batch mode. In incremental mode, the weights are updated after

each input is applied to the network, while in batch mode, all the inputs in the training set are applied to the network before the weights are updated.

To train an ANN, a set of experimental data is needed; these experimental data are divided into three subsets: the training set, the validation set and the test set. Training data are used to adjust weights and biases; then, the validation data are used to stop the training early if the network performance on the validation vectors fails to improve or remains the same. The test data, which remain occult to the ANN during the training process, are used to determine the ANN generalization capacity.

### 3.2. Construction of the ANN

To construct the ANN, the following steps were followed:

1. Collect data. To obtain a good predictor system, it is necessary to provide input data to train and test the neural network. Therefore, experimental measurements have to be made, which is costly. In this case, experimental data were obtained from a historical data basis, providing 429 data for training, testing and validating the ANN. The roughness of these 429 specimens was measured, as explained in Section 2.4, before (*Ra0*) and after being finished (*Ra*). However, as can be seen in Figure 2, the number of specimens for each angle is not the same. In the same figure, it is possible to appreciate the values of *Ra0* and *Ra* for each deposition angle.

2. Determine the input and output parameters. The ANN was designed to have nine input parameters: the above-mentioned two input parameters concerning specimen's characteristics: deposition angle (*anC*) and as-built surface roughness (*Ra0*), three blasting parameters: type of abrasive particles (*G1*), time (*T1*) and pressure (*P1*), and three electropolishing parameters: time (*T3*), voltage (*V*) and agitation frequency (*A*). For some specimens, before the electropolishing process, a second blasting was performed; however, in this case, the type and the pressure were kept constant and therefore only the time was considered as an input parameter (*T2*), which was set to zero for specimens of no additional blasting.

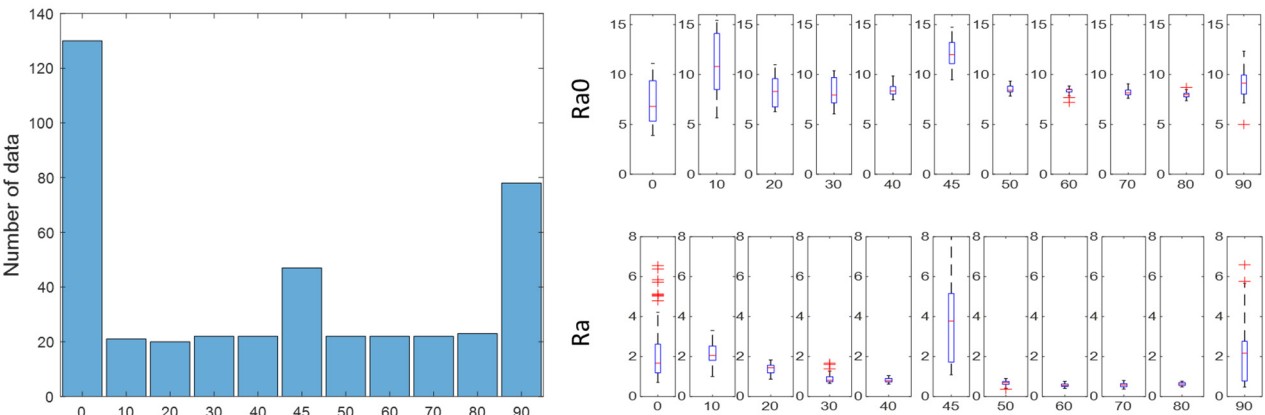

**Figure 2.** Number of specimens built with respect to the impression angle, and obtained values of *Ra0* and *Ra* for each angle.

It is worth mentioning that these data were not the result of an experimental design. Table 1 shows the parameters involved in the experimental data set and the values taken for each parameter. It is clear that each parameter did not have the same number of observed values (Figure 3), and nor did each observed value have the same number of elements (Figure 4).

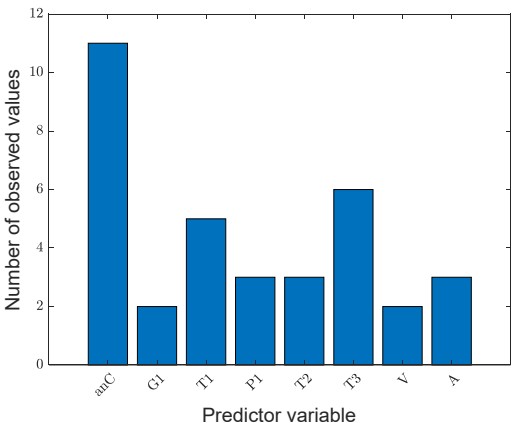

**Figure 3.** Number of observed values of each input parameter except as-built surface roughness.

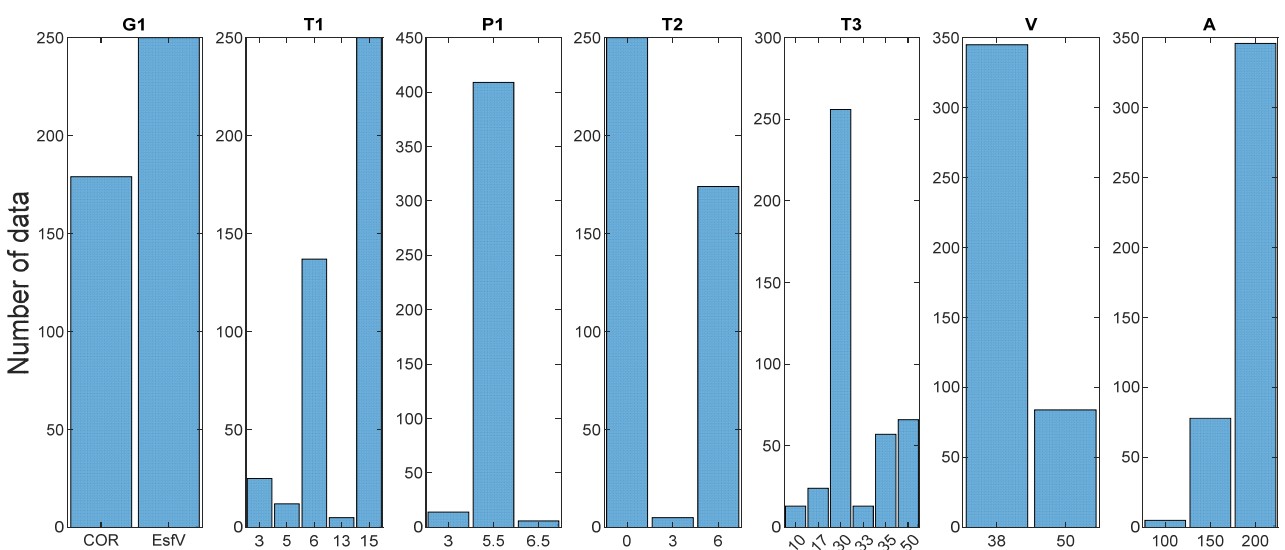

**Figure 4.** Number of data for each observed value.

The output parameter is the roughness of the surface after the polishing process (Ra). This means that the ANN structure should be 9-i-...-k−1, that is, the input layer had nine neurons, the output layer only one and there could be different numbers of hidden layers each one with different numbers of neurons (I, . . . , k).

3.  Scale the data set. As usual, to avoid numerical errors, all input data were normalized by rescaling experimental values to be between 0.1 and 0.9 using Equation (1) as is suggested in [27]:

$$x'_i = \frac{0.8}{x_{max} - x_{min}} + 0.1 \tag{1}$$

where $x'$ is the rescaled value, $x_{max}$ and $x_{min}$ are the extreme values of the corresponding parameter, and $x$ the original data. The qualitative variable (*G1*) was previously turned to quantitative.

4.  Determine the architecture of the network and the corresponding algorithm. *Matlab* was the software used to build, train and execute the neural network. The training function used to determine the weight and the bias of each neuron was the Levenberg–Marquardt backpropagation [31], which was implemented in batch mode. To apply an early stopping strategy [32], data were divided into three groups which were 80% of the data to train, 10% to validate and the rest for testing the network. The performance

function used in order to ensure the good behavior of the network was the mean squared normalized error (mse), and the transfer function was the *tansig* function (hyperbolic tangent). To decide on an appropriate architecture instead of following the guidelines given by [33], all architectures with one to three hidden layers of 1 to 20 neurons were considered, which supposed 8420 different architectures. It was decided to train all of them and select the one with the minimum mse. For each hidden layer, this procedure was repeated 25 times, obtaining as expected better results for architectures with three hidden layers. The minimum mse obtained was 0.317 which was achieved four times in all cases with the 9-6-10-19-1 network.

5. Test the system. It was found that the 9-6-10-19-1 network gave the best ANN model in predicting the value of surface roughness. Figure 5 shows the regression plot of each subset of the given dataset. A correlation factor of 0.918 was achieved when considering all data. It should be noted that the mse when considering test data was 0.22, showing that the ANN was not overfitting.

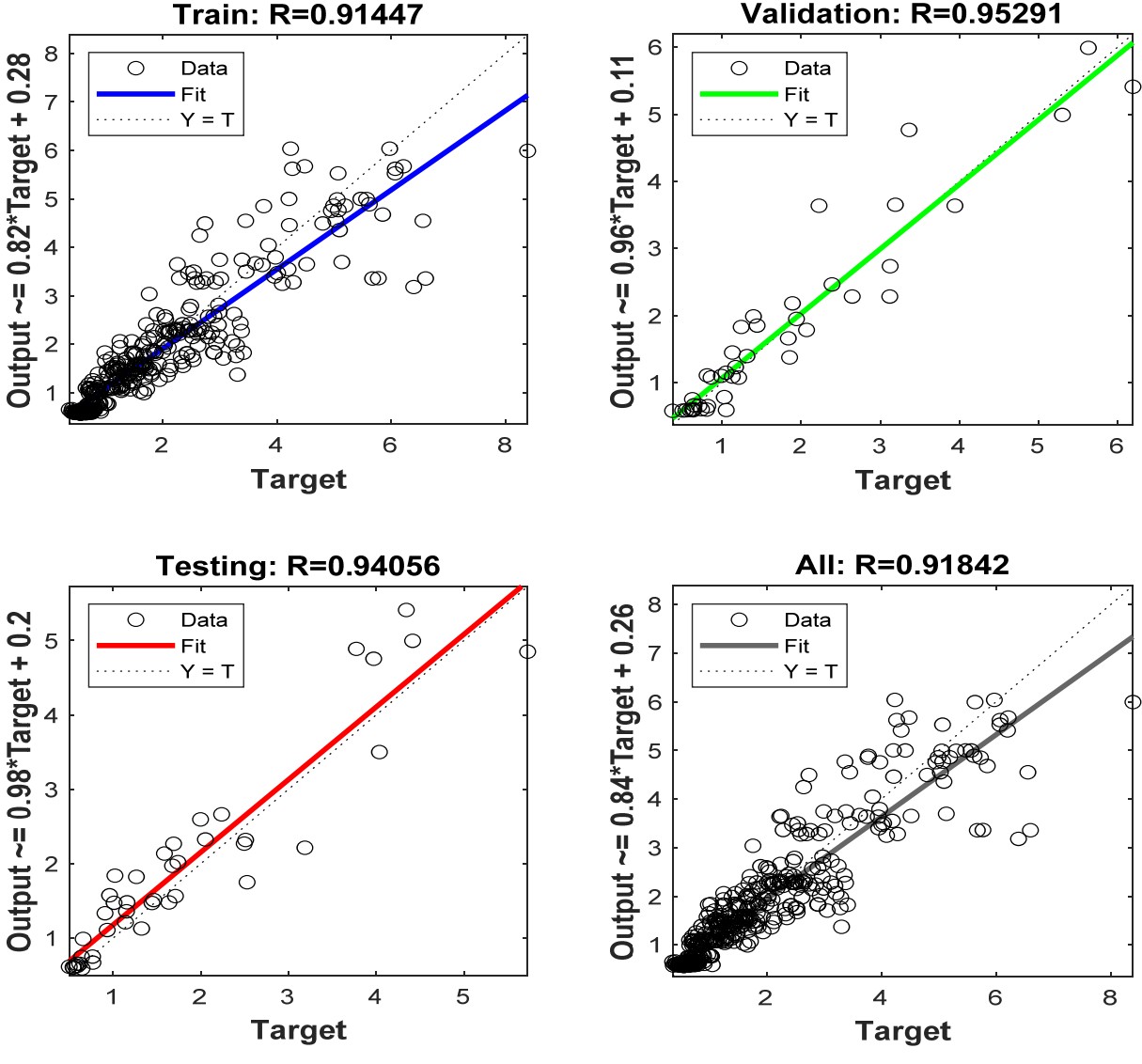

**Figure 5.** Regression plot of the ANN, which shows the relationship between the outputs of the network and the targets for each subset of the data.

## 4. Optimization Algorithms

As exposed in the precedent section, an ANN was built to predict the roughness of a flat surface of additively manufactured titanium alloy after being blasted and electropolished. However, this ANN provided an excellent tool that could be used in order to optimize both processes. The main idea was to develop an optimization algorithm using this ANN and the specimen characteristics (*anC* and *Ra0*) to obtain the blasting and electropolishing parameters that would ensure the lowest surface roughness. Of course, the *G1* parameter was fixed in advance.

Two optimization algorithms were consecutively used in order to obtain the optimum result. First a genetic algorithm (GA), in particular, the Matlab function *ga* [34] was used. Although these types of algorithms are time-consuming and not very accurate, they can be more efficient than traditional methods to locate the global minimum. Moreover, they cover a larger search space with the same number of operations [35]. In addition, they do not require a starting point to obtain the optimal parameters of the finishing processes to obtain a low surface roughness, and therefore, they are not being conditioned by the bias that could be introduced by the programmer.

The blasting and electropolishing conditions obtained after this first optimization were used as the starting point of a nonlinear optimization algorithm. In this case, Matlab's *fmincon* [36] algorithm was used, which is faster and more accurate than the *ga*. This function is based on a sequential quadratic programming method, which mimics Newton's method for constraint optimization. It is an iterative method of starting from some initial point and converging to a constrained local minimum [37]. At each iteration, it solves a quadratic program that models the original nonlinear constrained problem at the current point. The obtained solution is used as a search direction to find an improving point, which is the next iteration. The iteration is repeated until an optimal solution is found. However, it was proven that in the present case, the given solution was strongly dependent on the starting point. This is why a previous optimization with a genetic algorithm was carried out.

It is worth mentioning that both algorithms were limited to working in the interpolation zone of the involved parameters.

## 5. Results and Discussion

Figure 5 shows the as-built surface roughness (*Ra0*) and after the surface treatments. (*Ra*). Although there was an obvious improvement in the surface roughness, the mean value of *Ra0* was 8.58 μm, while *Ra*'s was 1.81 μm, a huge dispersion in surface roughness can be observed; the standard deviation of *Ra0* and *Ra* was 2.21 and 1.42 μm, respectively, which suppose a coefficient of variation of 25.5% for *Ra0* and 78.6% for *Ra*. In the right-hand side chart of this figure, it can be observed that there is also a high dispersion in surface roughness versus angle, especially at angles with a greater number of specimens (0°, 45° and 90°). The solid line corresponds to a third-degree polynomial that approximately fits the data, this suggests a slight correlation between surface roughness and the deposition angle.

On the one hand, *Ra0* dispersion could be explained by several factors that are not controlled in this work, those concerning the manufacturing process, as could the variations in the laser power, the speed of deposition, or the quality of the Ti6Al4V titanium alloy powder. On the other hand, the dispersion in *Ra* is explained by three main factors: Firstly, obviously, the dispersion in *Ra0*. It is clear, by observing the right-hand side chart of Figure 6, that a high dispersion in *Ra0* of angles 0°, 10°, 45° and 90° implies, likewise, a high dispersion in *Ra* values. Secondly, in the case of angles of 0°, 45° and 90°, the fact must be added that both blasting and electropolishing techniques were applied on the basis of the prior knowledge of the manufacturer, knowledge that was based on a trial and error process, of course, the fact that more specimens were available for these angles, allowed greater variation in the way the finishing processes were applied, resulting in a higher dispersion on Ra. Thirdly, and as a complement to this explanation, it was pointed out in Section 2 that some specimens were blasted two times, and the case is that only

specimens of these three angles were affected by this extra blasting process which yielded the worst results.

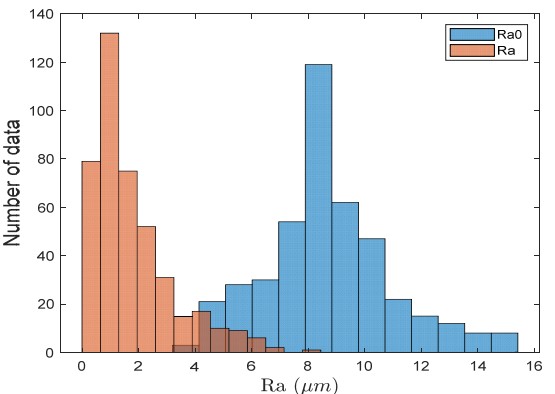 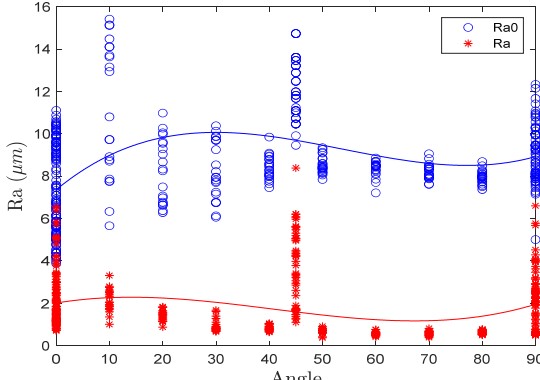

**Figure 6.** Huge dispersion was observed in surface roughness before and after the surface treatment.

The existence of developed ANN gives those manufacturers the option to virtually test these variations in the blasting and electropolishing processes, moreover, it was implemented in a friendly user interface (Figure 7) using Matlab.

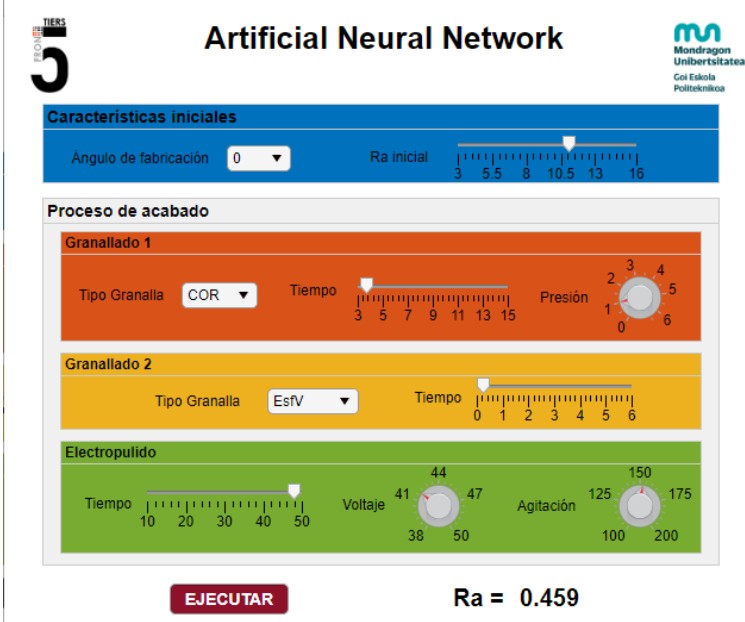

**Figure 7.** User interface to be used to virtually test variations in blasting and electropolishing applications.

Finally, the combination of the ANN and the optimization algorithms allows the possibility of significantly improving the surface finishing. In Figure 8, the mean value of *Ra0* and *Ra* for each angle when glass microspheres were used for a single blasting process can be observed, note that 45° is missing because, for this angle, all the specimens were blasted two times. The *Ra0* values were introduced in the optimization algorithm to obtain the expected optimum values, values which are also given in the bar diagram. The obtained mean value of *Ra* was 0.97 μm, while the predicted value was 0.38 μm, which suppose a significant improvement of the surface roughness as this was a change of 60.8%. Moreover, the dependence on as-built surface roughness is much less clear.

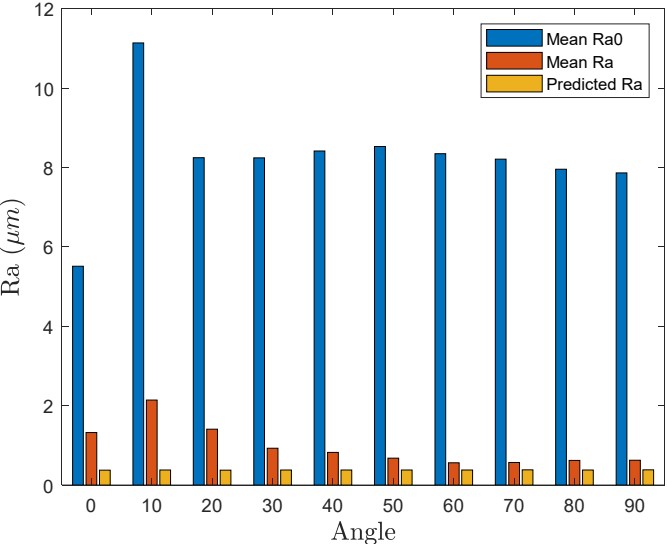

**Figure 8.** Mean values of surface roughness as-built (Ra0) and after blasting and electropolishing processes for each angle. The predicted values are in yellow after being applied to the conditions given by optimization algorithms.

When the optimization is finished, with the aim of evaluating the influence that each parameter had on the optimal Ra, the resulting Ra when each parameter was moved along the definition interval maintaining the rest of them fixed in the optimal position was computed. The optimization result depends on the initial conditions: *Ra0*, the deposition angle and the type of blasting particle. As an example, Figure 9 shows the obtained curves for the case of 80°, *Ra0* = 7.9 μm and using glass microspheres. Therefore, it can be observed that the most influencing parameters on the final roughness are *T3* and *P1*, that is, electropolishing time and pressure of the blasting process. The same trend was observed for other initial conditions.

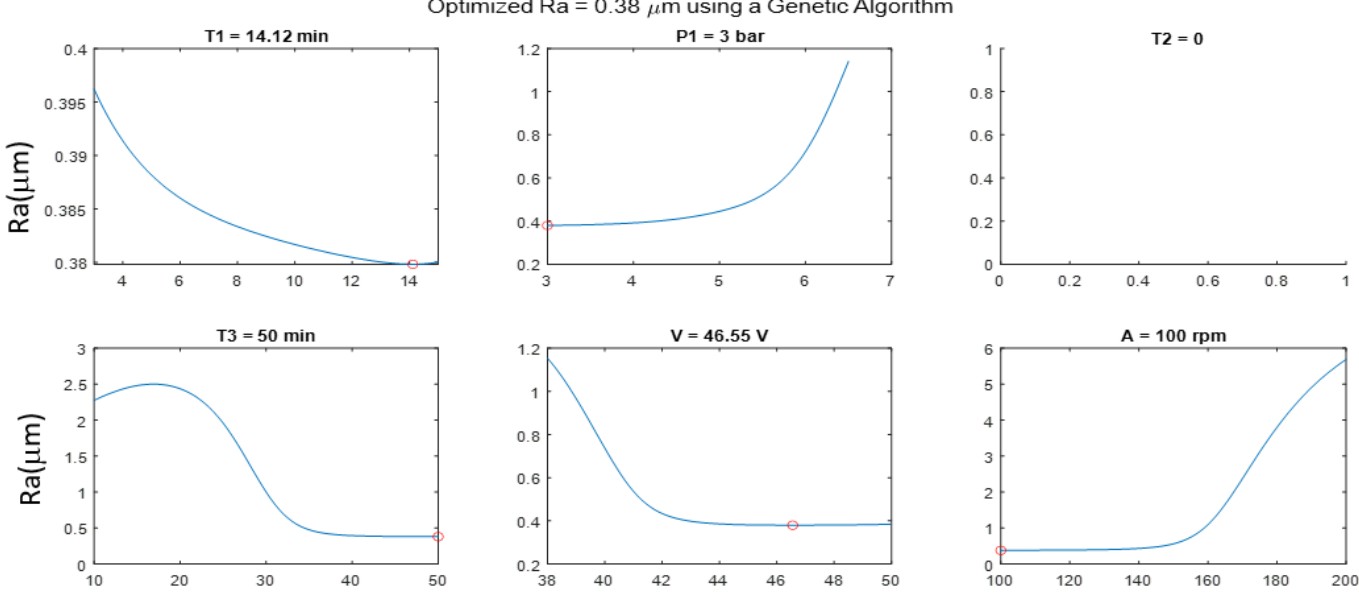

**Figure 9.** "Sensivilty" curves for each ANN parameter when deposition angle was 80° and *Ra0* 7.9 μm.

When observing the concrete conditions that should be applied in blasting and in electropolishing processes, it was appreciated that, in some cases, certain parameters were

placed at an extreme of their definition interval, suggesting that maybe better conditions could be obtained if these intervals were enlarged. Taking into account that extrapolation is always risky, this offers the opportunity to carry out a new collection of tests to improve the knowledge of the process.

It is worth mentioning that the extra blasting process yields worse results when focusing on roughness behavior. In the case of 0 and 90°, where specimens were treated with and without extra blasting process, the *Ra* obtained was 2.61 μm and 1.32 μm, respectively, for horizontal specimens, while for vertical specimens, the obtained values were 2.75 μm and 0.62 μm. This behavior was logically reproduced by the ANN; however, the obtained optimal results were much more similar, being 0.377 μm and 0.375 μm for angle 0° and 0.440 μm and 0.385 μm for the case of 90°.

## 6. Conclusions

In this paper, an ANN to predict the surface roughness of additively manufactured Ti6Al4V alloy specimens after being blasted and electropolished is developed. This ANN is then used to determine the parameters of both finishing techniques in order to obtain a softer surface. The following outcomes can be drawn from the results presented above:

- The ANN is a powerful tool to virtually perform blasting and electropolishing tests, as it has a correlation factor greater than 0.9.
- The optimization algorithm gives the conditions to be applied to improve, roughly by 60%, the surface roughness.
- The analysis of issues given by the ANN allows for determining the most influential parameters and suggests new tests to be carried out to obtain better results. These tests could also be used to improve the ANN itself.

**Author Contributions:** D.S.: Writing—Original draft preparation, Conceptualization, Methodology, Software, Validation, Formal analysis. M.T.: Software, Validation, Data Curation, Writing—Review and Editing. M.B.G.-B.: Conceptualization, Supervision, Resources, Writing—Review and Editing E.E.: Data Curation, Resources, Investigation, Writing—Review and Editing. M.C.: Investigation, Writing—Review and Editing, Visualization. P.J.A.: Conceptualization, Supervision, Resources, Writing—Review and Editing, Visualization. All authors have read and agreed to the published version of the manuscript.

**Funding:** This work was partially supported by the Departamento de Desarrollo Económico y Competitividad of the Basque Government (ELKARTEK 2019 KK-2019/00077) and by the Ministerio de Ciencia e Innovación of the Spanish Government (project SURF-ERA, EXP—00137314/CER-20191003).

**Acknowledgments:** The authors want to kindly acknowledge LORTEK for providing the SLM samples.

**Conflicts of Interest:** The authors declare no conflict of interest.

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
