# Peer review of "Prediction of Surface Roughness of SLM Built Parts after Finishing Processes Using an Artificial Neural Network"

_jmmp, doi:10.3390/jmmp6040082_

Round 1

Reviewer 1 Report

The manuscript presents the applicability of Artificial Intelligence algorithms for the prediction for surface roughness of SLM built parts. Although, the authors have carried out a detailed study of the applicability of ANN the manuscript has many weak points.  An issue is the quality of presentation which is rather poor and makes difficult the comprehension of the topic. The main problem is that the manuscript presents no results of the training and test phase beside a value of correlation coefficient. Therefore additional information should be included in the manuscript to support the conclusions. Additionally, some further issues (see below) make the publication in its current form rather difficult. 

- Introduction: ''Numerous publications ...''. The authors should either rephrase the word numerous or provide more references for the connection of the building angle with surface roughness and mechanical properties.

- Section 2 first sentence: ''..an ANN is an AI ...''. The authors should provide the term Artificial Intelligence and after that they can use the abbrevation.

- Section 3.1: ''...which is a computational systems...'' systems -> system.

- Section 3.1 last paragraph. Is the validation set also used in the training phase or is just used to avoid an unnecessary long training phase when the error is large? The authors should provide a more detailed description of the training method.

- Section 3.2. The authors mention that the origin of the experimental data is a historical data basis. Are the data the same as that obtained from the tests described in section 2? If the source of the training data is the literature the authors should refer to it.

- Figure 3 and corresponding test (line 192). The should add a clarification that the levels are actually showing the number of the observed values for each of the input parameter.

- point 3 scale the data: '...'using the following expression [23]'' this phrase would be better as ''...using the equation 1 as it is suggested in [23].''

- point 4 architecture: ...all of them were trained 25 times..'' Do you mean with 25 different training sets? In that case what was the scatter among the different performances?

- point 5 test the system: is the correlation factor of 0.92 on the test set? The authors should present the results of the training and test phase as single plots observation vs prediction with the line y=x as a reference of a flawless performance.

- Section 5. The authors should consider to show additional to the standard deviation the coefficient of variation of Ra0 and Ra which allows a quick understanding of the scatter degree.

- Figure 5 Right chart: Are the solid lines the predictions of the ANN algorithm? Also the caption should be ''... roughness before and after the surface treatments.''

- Line 258: ''..especially at angles with a greater number of specimens (0, 45, 90).'' Do you mean the dispersion for each Ra0 and Ra? In that case at 10 degrees the Ra0 shows the largest scatter. Additionally, from the results it seems that Ra, Ra0 are somehow amplified at 0, 45 and 90 C. Do you have any explanation for this?  

- Figure 6. Is the user interface implemented in Matlab?

- Figure 7. The caption should change from Figure 1 to 7. Additionally, is the test, the training or the entire dataset presented in the figure and why is the data of 45o missing? Furtermore, from the presentation of the predicted Ra it appears that it is constant along the different angle. Also the predicted values do not seem to be in accordance with the mean Ra values. 

- Figure 8. Change caption from Figure 2 to 8. Addtionally, is there a reason to present an empty chart (T2=0)? The charts should have labels on their axis.

- Conclusions. First bulletin ''correlation factor greater than 0.9''. This is showed nowhere in the manuscript. The authors should present the results of training and test sets (see also comment above).

- Conclusions: Second bulletin. Could this 60% change if all the different training processes considered. In that case a mean value Ra as a result 25 (or xxx ) itterration of ANN algorithm would be more representative since it would include also the error due to different training set.

Author Response

We have responded to the reviewer in the attached file.

Reviewer 2 Report

This paper demonstrates the surface roughness prediction of SLM built parts after finishing processes using ANN. The work is interesting and the research work is quite completed. However, the innovation of this work should be further elaborated. The following concerns should be addressed.

1.Why use ANN? Should be explained.

2.Literature review should be improved. Literature review on the prediction of surface roughness using AI Method is needed. There have been many AI based methods for the surface roughness prediction. I think this ref (R. Wang, et al. Ensemble learning with a genetic algorithm for surface roughness prediction in multi-jet polishing. Expert Systems with Applications. 2022 , 118024.) can help.

3.The fontsize of all figures can be enlarged. Enlarge Fig. 6.

4.The number of Fig. 7 and 8 are not right. Please check the whle paper carefully.

5.The obtained mean value of Ra was 0.97um, while the predicted value was 0.38um. You mentioned this represents an improvement of 60.8%. Could you further explain this. Normally, this is the deviation between the practical value and predicted value.

6.Do you think what is the advantage of ANN method mentioned in this paper?

Author Response

(The authors gave the same response as above.)

Round 2

Reviewer 1 Report

The authors have provided all necessary corrections and clarifications. No further comments.

Reviewer 2 Report

I have no more comments.